# Applications and Limitations of Neuro-Monitoring in Paediatric Anaesthesia and Intravenous Anaesthesia: A Narrative Review

**DOI:** 10.3390/jcm10122639

**Published:** 2021-06-15

**Authors:** Chiara Grasso, Vanessa Marchesini, Nicola Disma

**Affiliations:** 1Unit for Research & Innovation, Department of Paediatric Anaesthesia, IRCCS Istituto Giannina Gaslini, 16147 Genova, Italy; chiara.grasso210@gmail.com; 2Paediatric Intensive Care Unit, Royal Melbourne Children’s Hospital, Parkville 3052, Australia; vane.marchesini@gmail.com

**Keywords:** neuromonitoring, EEG-derived monitor, depth of anaesthesia, near-infrared-spectroscopy, brain oxygenation, anaesthesia, paediatric

## Abstract

Safe management of anaesthesia in children has been one of the top areas of research over the last decade. After the large volume of articles which focused on the putative neurotoxic effect of anaesthetic agents on the developing brain, the attention and research efforts shifted toward prevention and treatment of critical events and the importance of peri-anaesthetic haemodynamic stability to prevent negative neurological outcomes. Safetots.org is an international initiative aiming at raising the attention on the relevance of a high-quality anaesthesia in children undergoing surgical and non-surgical procedures to guarantee a favourable outcome. Children might experience hemodynamic instability for many reasons, and how the range of normality within brain autoregulation is maintained is still unknown. Neuro-monitoring can guide anaesthesia providers in delivering optimal anaesthetic drugs dosages and also correcting underling conditions that can negatively affect the neurological outcome. In particular, it is referred to EEG-based monitoring and monitoring for brain oxygenation.

The present narrative review aims at presenting current available monitoring of depth of anaesthesia and brain oxygenation and critically discussing their application in the paediatric clinical setting. 

## 1. EEG-Based Anaesthesia Depth Monitoring

The monitoring of hypnosis depth by means of electroencephalogram-based (EEG-based) systems is emerging in paediatric anaesthesia common practice. This monitor system measures specific EEG signs which derive from anaesthetic-induced changes in neuronal firing.

In children, these signs are influenced not only by both the depth of sedation and the mechanism of action of the specific anaesthetic agent, but also by the typical age-related changes of a developing brain [1]. Moreover, the current clinical condition and underlying co-morbidities might significantly affect the validation and application of this monitoring [2,3,4]. Therefore, although recent studies have shown that they are worth being used in children [5], many of the available monitoring methods show limitations when applied to paediatric patients and have to be cautiously used, especially in those aged less than 1 year [6,7,8]. 

### 1.1. Available EEG_S_ Technology 

For daily practice, in addition to the unprocessed EEG (the raw cortical EEG), two different groups of EEG-based devices are currently available. These are the EEG-derived indices (processed EEG or p-EEG) and the spectrogram or Density Spectral Array (DSA) [9] (Figure 1).

The majority of the EEG-derived monitors works by disassembling a complex EEG waveform (the raw EEG), into many series of waves of different frequencies. These are, respectively, the slow (<1 Hz), delta (1–4 Hz), theta (4–8 Hz), alpha (8–13 Hz), beta (13–25 Hz) and gamma (25–40 Hz) oscillations. The power of each band is defined by quantity and amplitude [9].

The processed EEG (p-EEG) devices convert several of the EEG variables into a single index through mathematical algorithms. The obtained index represents the level of hypnosis. Some examples of p-EEG monitors are the Bispectral Index Monitor (BIS, Medtronic Inc.), the Patient State Index (PSI, Masimo Inc.) of the SEDLine brain function monitor or the Narcotrend index (Narcotrend Inc.). The Spectral Edge Frequency (SEF95) is a common EEG-processed parameter, which represents the frequency containing 95% of the power. Finally, the Response and the State Entropy measures the degree of disorder in the EEG together with the electromyography (EMG) signals [10]. 

Differently from the pEEG, the Density Spectral Array (DSA) or spectrogram is a real time monitoring method which portrays all the EEG frequencies and their power over the time in a three-dimensional method. The latter is then integrated in a two-dimensional plot using colours to represent different powers [11].

Table 1 lists some of the most widely used EEG-based anaesthesia monitoring devices which are currently available, along with their key features.

### 1.2. Clinical Application

The EEG ranges from a wakefulness pattern characterised by beta (13–25 Hz) and gamma (25–40 Hz) oscillations, to an isoelectric pattern, which represents an anaesthetic-induced coma with profound hypothermia [9]. An unconscious brain under surgical anaesthesia usually shows slow (<1 Hz) and alpha (8–13 Hz) oscillation. After induction of anaesthesia or in case of excessive sedation, slow oscillation or occasional burst suppression can occur. However, the EEG pattern also differs in relation to the anaesthetic drug administered. In this way, the DSA allows a trained anaesthetist to easily identify the specific patterns which correspond to certain hypnosis depth and anaesthesia trends [9,12]. 

Since the EEG-based devices correlate with the amount of both volatile and intravenous anaesthesia, they can predict the patient’s state of consciousness or unconsciousness [7]. They should lead to a better control of anaesthesia depth which, in turn, allows avoiding both over-sedation and light-anaesthesia [13,14]. In adults, this has been proven not only to reduce time to emergence, tracheal tube removal, and time to PACU discharge but also in decreasing post-operative nausea and vomiting and delirium as well as in lowering the overall amount of anaesthesia given. [15,16,17].

In children, only a few studies have analysed correlations between EEG-based monitors and the post-anaesthesia outcome [18,19,20,21]. Moreover, the currently available devices for the paediatric population are all based on algorithms which have been tested on adults [7].

Notwithstanding, some studies demonstrate that BIS and Narcotrend monitors are associated with a reduced emergence time and a decreased discharge-time from PACU after paediatric general anaesthesia [18,19]. However, discordant results have been found in a randomised trial on the same topic in paediatric cardiac patients [20]. Although a correlation between the use of pEEG devices and the reduction of awareness episodes has not been clearly demonstrated in children, the anaesthesia monitoring is considered to be also useful in this area [21]. An ongoing prospective, observational, multicentre study aims to further investigate the role of EEG monitoring in infants younger than 36 months. The study observes the incidence of isoelectric EEG events (defined as EEG amplitude less than 20 μV for 2 s or more) and its association with neurological outcome and quality of life in children undergoing at least 30 min of inhalational or propofol-based general anaesthesia [22].

Despite some studies demonstrated that EEG-based monitoring devices appropriately correlate with specific EEG signs during sevoflurane and propofol anaesthesia in children older than 1 year of age [5,13], the paramount evidence of their beneficial effects is mostly related to total intravenous anaesthesia and propofol (TIVA). In fact, a linear slowing of EEG and a lowering of BIS index has been demonstrated concurrently with the increasing in propofol blood concentration in children [5].

A survey of European paediatric anaesthetist in 2018 showed that the main application of EEG monitoring occurs during TIVA and in children over 4 years of age, with the aim of avoiding awareness [23]. Furthermore, the use of TIVA has recently emerged into the common practice of paediatric anaesthesia. However, despite fairly reliable models, doubts still exist regarding the pharmacokinetic and pharmacodynamic variability of anaesthetic agents among different ages. In this scenario, the use of anaesthesia depth monitoring systems results very useful, especially when the TIVA is associated with Neuromuscular Blockers (NMBs) [24]

### 1.3. Limitations of pEEG and DSA in Children

In spite of the fact that they are taking hold also across paediatric anaesthesia standard practice, EEG-based monitors have some limitations, especially in infants (1–12 months) and neonates (<1 months).

Age. The pEEG devices in current use rely on data from adult patients. Thus, their suitability to monitor anaesthesia depth in children is not well-defined, as the EEG changes accordingly with brain development. Only around the age of 12 years, the brain reaches its maturity as well as the stabilization of its electrical activity [25,26,27,28]. These age-related changes in the EEG, which mirror the synaptogenesis and myelination during the first year of life, also diminish the total DSA power and its representativeness of the depth of hypnosis in infants to the point that the devices currently in use cannot be deemed accurate and reliable in paediatric patients less than 6 months of age [1,11,29,30].

Drugs. Anaesthetic agents affect the EEG-derived depth monitors resulting in a binary and non-specific modification of the index. Wang’s study on single propofol anaesthesia in children, showed a linear correlation between propofol concentration and BIS value [26]. Conversely, those on inhalation anaesthesia with sevoflurane demonstrated a decrease in BIS value when the concentration of sevoflurane increased from 0 to 4%, but a paradoxical increase when the volatile anaesthetic rose to values above 4%. An increase in BIS and entropy was also observed during ketamine anaesthesia. This was paradoxically linked to a deeper hypnosis [31]. Dexmedetomidine leads to a decrease in the BIS value when used together with sevoflurane or propofol to induce general anaesthesia [32,33] while discordant data derive from studies on N_2_O and pEEG monitors [34,35]. Neuromuscular blockers (NMBs) lead to a rapid decrease in BIS in adults. This probably happens due to the impact of NMBs on EMG activity [36]. Finally, whether opioids affect BIS is neither well-defined nor consistent [37,38]. However, this could be explained by the fact that the nociception neuronal pathway is not well-detected through cortical EEG-based monitors [7]. Raw EEG and spectrogram analysis, instead, allow an experienced physician to recognize the specific effects that each anaesthetic has on the brain’s cortical electrical activity. This usually reflects the drug’s mechanism of action [29].

Neurological Diseases. Caution should be paid when EEG-derived monitors are applied in children with neurological diseases such as congenital metabolic and genetic disorders and acquired diseases, such as post-hypoxic encephalopathy or neurodegenerative disorders of unknown origin because lower BIS values and a greater tendency to burst suppression at comparable doses of anaesthetic has been described in these patients [2,3,4]. This is likely related to epileptic and non-epileptic EEG anomalies caused by the underlying neurological disorder [2]; however, the same effect may also be due to the use of drugs that act on the nervous system such as antiepileptics or neuroleptics. Nonetheless, not only can the EEG monitors be used, but also they are extremely useful in those patients whose communication problems and severe underlying neurological impairment can make it difficult to assess an adequate level of anaesthesia [3].

## 2. Near Infrared Spectroscopy (NIRS) in Children

One of the goals of good intraoperative care is to maintain adequate cardiac output and delivery of oxygen to the tissues, particular to the brain. Monitoring of the standard vital signs provides information about end-organ perfusion, but lacks specificity for brain perfusion. Blood pressure measurements during anaesthesia are often used as a proxy for organ perfusion, but it has been found inaccurate, especially in neonates and infants [39]. We know that hypotension is one of the perioperative risks related to neurocognitive developmental issues in this population [40]. However, the way cerebral autoregulation is maintained is multifactorial. Arterial blood pressure is only one of the contributing factors and it cannot be used as a surrogate for organ perfusion in general and specifically for brain perfusion.

Real-time near infrared spectroscopy (NIRS) noninvasively measures cerebral oxygen saturation (ScO_2_) in the tissue approximately 1 to 2 cm below the sensor. This method is based on the high degree of transparency of brain tissue in the near-infrared range. Even though NIRS reads haemoglobin saturation in all the vessels located in the portion of the tissue scanned (arterioles, venules, and capillaries), it is a surrogate of venous saturation because venous volume counts for about 75% to 85% of the total cerebral blood volume and only 15% to 25% of it is arterial. Cerebral oximetry using NIRS technology has been validated against jugular venous oxygen saturations, even in children with congenital heart disease, with a correlation coefficient between jugular venous bulb saturations and cerebral oximetry as high as 0.8 to 0.9 in some studies but as low as 0.5 in others [41,42].

### 2.1. Population Target

The application of cerebral NIRS in children under anaesthesia has not yet become a clinical standard of care, even though a growing body of evidence suggests that NIRS can be very useful in detecting cerebral desaturation in particular population and for specific type of surgery. NIRS can be a valuable tool in neonates and infants as they are more vulnerable to neurological complications related to disruption of cerebral homeostasis [43]. In particular, preterm infants have anatomically incomplete and underdeveloped cerebral vasculature that cannot yet autoregulate fully [44]. Some studies of cerebral autoregulation in neonates point out a correlation between impaired cerebral autoregulation and clinical outcome [45,46]. Therefore, it might be crucial to be able to adequately and safely assess the autoregulatory capacity and to detect and prevent unnoticed cerebral hypo- or hyper-perfusion and secondary brain injury. NIRS might fulfil these requirements and can be used to measure regional cerebral oxygenation as a surrogate for disruption of autoregulation.

It is also known that cardiac surgery and cardiopulmonary bypass itself are associated with dysfunctional autoregulation [47]. In this context, fluctuation of ScO_2_ from baseline may represent an imbalance between oxygen supply and demand. It has been shown that these changes occur before lactic acidosis or hemodynamic compromise are established [48,49]. This has been suggested based on findings that correlate occurrence and duration of cerebral desaturation intraoperatively with low cardiac output state and elevated lactate levels as well as increased mortality [50,51,52]. Moreover, NIRS works effectively also in non-pulsatile circumstances, like during CPB, other low-flow states and during hypothermic circulatory arrest [53]. 

Despite numerous studies have shown that low ScO_2_ can anticipate later signs of imbalanced oxygen delivery/consumption, the long-term benefit of this approach has yet to be elucidated. A 2009 systematic review of cerebral NIRS in paediatric cardiac surgical patients including 38 trials found no evidence that cerebral NIRS monitoring improved patient outcomes [54]. Since this review, no new studies have been conducted that demonstrate that changes in cerebral oximetry prompt interventions that improve clinical outcomes. This is likely to be related with lack of clarity on the threshold value below which risks increase. 

Evidence supporting the benefit of NIRS monitoring in noncardiac surgical procedures are still very discordant. On one hand, the majority of papers delineating the use of NIRS monitoring during major noncardiac paediatric surgery have been published in surgical journals. In these studies, NIRS was mainly used as a monitoring tool to address possible safety issues of new surgical techniques, such as thoracoscopic repair of congenital diaphragmatic hernia or oesophageal atresia [55,56,57]. On the other hand, the use of NIRS has been triggered by the issue of anaesthesia neurotoxicity in children less than 3 years of age. Early studies have been conducted with the aim of identifying incidence and associated factors of cerebral desaturation in children undergoing anaesthesia for non-cardiac surgery and therefore see if any undetectable cerebral desaturation could be a risk factor in the anaesthesia-related neurotoxicity. It has been shown that in children less than 2 years of age, ScO_2_ usually increased with sevoflurane induction of anaesthesia. Nonetheless, when significant hypotension occurred at induction, ScO_2_ decreased, and this happened mainly in younger children [58,59,60]. These studies suggest that unrecognized cerebral desaturation during anaesthesia of infants occurs frequently and is often associated with hypotension. Two more recent studies have found different results. Olbrecht et al. [61] described the incidence of cerebral desaturation in 453 infants less than 6 months of age undergoing a wide range of non-cardiac procedures. Severe desaturation (below 50% absolute value or greater than 30% below baseline) was present in 2% of the population observed, but it was poorly associated with low mean arterial pressure. Their conclusion was that unrecognized severe cerebral desaturation is uncommon. Gomez-Pesquera et al. [62] investigated the relationship between cerebral oxygenation and negative postoperative behavioural changes (NPOBC) 7 days after surgery in children between 2 and 12 years of age. They showed an association between decreased cerebral oxygenation and the incidence of NPOBC, though not for patients experiencing the biggest desaturations (>20% below baseline).

### 2.2. ScO_2_ Values and Trigger for Intervention

When using NIRS technology, it is important to understand the relevant physiological parameters contributing to cerebral regional oxygen saturation, which are heart rate, blood pressure, oxygen saturation, and carbon dioxide partial pressure Figure 2. All these variables need to be considered during interpretation of NIRS and its changes can only be interpreted together with standard vital signs monitoring. In a recent editorial, Skowno et al. referred to NIRS as a “multidimensional monitor” because it is influenced by multiple variables and its understanding relies on these variables [63]. Imbalance between oxygen demand and supply as well as ventilation can be detected early in the context of changes in cerebral NIRS while standard monitoring parameters can help identifying the causes of ScO_2_ changes. Therefore, interventions can be made in a timely matter in order to prevent potentially life-threatening complications or at least reduce their severity. It is important to highlight that a decrease in ScO_2_ can be also due to low haemoglobin concentrations following acute blood loss even before haemodynamic instability occurs. Anaemia becomes relevant for ScO_2_ when haemoglobin levels are not adequate to deliver an oxygen-carrying capacity that meets the demands of cerebral oxygen consumption. Transfusion of red blood cells has been shown to result in increased ScO_2_ values [64].

Values of ScO_2_ within normal range vary in relationship with patient population and NIRS instrumentation, with high interindividual and intraindividual variability. Mean (±standard deviation) values for ScO_2_ in healthy children are 68% ± 10% and in infants 64% ± 5% [65,66]. Determination of what constitutes a critical ScO_2_ value depends on the identification of a hypoxic-ischemic threshold, beyond which ongoing hypoxia and/or ischemia leads to neurophysiologic impairment, cerebral metabolic failure, and irreversible morphologic damage. It is unknown what the exact individual lower safety margin of ScO_2_ values in children is. Data from animal studies suggest that the severity and the duration of low cerebral oxygenation are the main leading factors to cerebral damage [67,68].

### 2.3. ScO_2_ and Clinical Algorithms

Despite scientific efforts, it has not yet been possible to define individual lower limit of ScO_2_ values. This uncertainty applies in an anaesthetic clinical setting, where it is difficult to identify a ScO_2_ value that should trigger intervention. Because of it, anaesthetists base their approach on personal experiences.

Few algorithms have been published in order to guide the NIRS use, interpretation and intervention. They are all based not on scientific evidence, but on practical approaches led by physiological and pathological knowledge and common sense. Kurth et al. consider 60% to 80% as a normal range for ScO_2_, and because functional impairment begins at an ScO_2_ of about 45% (albeit in piglets), there is a buffer zone between 45% and 60% whereby cerebral oxygenation is adequate for function, but lower than normal [68]. Decreasing ScO_2_ and/or absolute values below the buffer zone necessitates prompt re-evaluation of the patient.

Denault et al. [69] published a proposed clinical algorithm to respond to changes in cerebral oximetry, using clinical parameters including blood pressure, arterial saturation, carbon dioxide tension and haemoglobin. The algorithm has been developed for adult patients, and its use cannot be validated in the paediatric population because there are insufficient data to support this interventional algorithm in this population.

Weber F et al. developed the “Baseline-Bottomline Approach” as NIRS-directed haemodynamic management [70]. It differs fundamentally from all previously published approaches, as it defines a baseline ScO_2_value, registered in the awake child prior to anaesthesia induction, as the lowest acceptable limit during anaesthesia and surgery. The strategy to ensure adequate cerebral perfusion and oxygen delivery uses cerebral SO_2_ as the single target parameter, while blood pressure, heart rate, PaCO_2_, and SaO_2_ are contributing parameters that determine the ScO_2_. The aim of this approach is to maintain ScO_2_ at or above awake baseline, adjusting and correcting the contributing variables. Basically, they aim at maintenance of an optimal condition rather than treatment of potentially harmful cerebral desaturation.

## 3. Conclusions

EEG-derived anaesthesia depth monitors perform well in children that should be applied for guiding clinicians in providing the correct dose of general anaesthetics. However, clinicians should be trained on the specific device applied with its own technology, in order to draw the right interpretation. Moreover, age, type of anaesthesia and underlying neurologic conditions should guide clinicians to correctly titrate medications, beyond the monitoring result by itself.

NIRS and brain oxygenations monitoring represent a relatively recent technological advancement. It is clear that acute drop in brain oxygenation will have an impact on the neurological outcome, but the specific intervention that should be applied for acute derangements is still to be validated. NIRS can monitor brain oxygenation in the area where the sensor is applied, and it should not be interpreted as a surrogate index of brain autoregulation. In fact, the limits of autoregulation may vary according to the age, but also to the other physiological parameters. Then, NIRS should be clinically interpreted under the light a more comprehensive clinical condition, which might evolve time by time during anaesthesia and surgery. Nevertheless, both monitoring should be implemented in the daily practice, especially when the most fragile population is treated. Understanding the physiology and pathophysiology of brain autoregulation will be the next step of clinical research to define the limits of tolerance. Right now, clinicians should bear in mind that keeping young patients as close as possible to the baseline parameters is mandatory to stay away from the edge of loss of autoregulation.

## Figures and Tables

**Figure 1 jcm-10-02639-f001:**
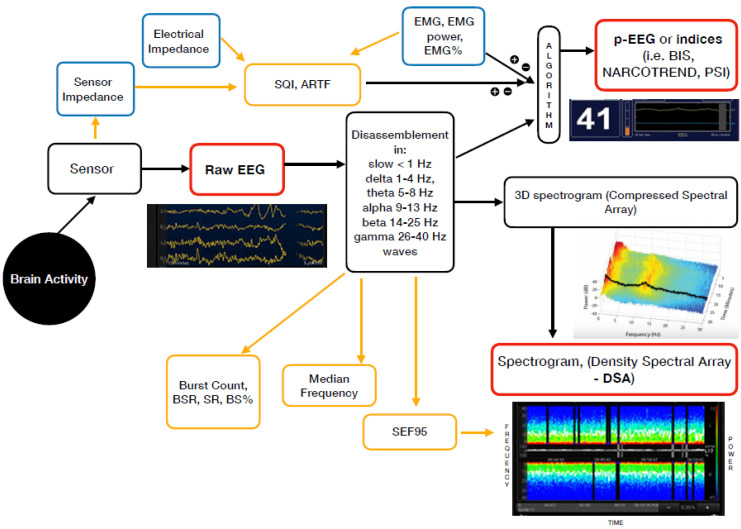
Mechanism of action of the currently available EEG derived anaesthesia monitoring (in red). The sensor is placed on the forehead and records the electrical potentials produced in the cortex resulting in the raw EEG. The majority of the EEG-derived monitors work by disassembling a complex EEG waveform (the raw EEG), into many series of waves of different frequencies (Hertz—Hz). These, together with several of the EEG variables, are converted into a single index through mathematical algorithms by the processed EEG (p-EEG) devices. The obtained index represents the level of hypnosis. However, the index can also be affected by the Electromyography (EMG) and the Signal Quality Index (SQI or ARTF). The spectrogram, is a real time monitoring which portrays all the EEG frequencies and their power over the time in a three-dimensional method (3D spectrogram or Compressed Spectral Array). The latter is then integrated in a two-dimensional plot using colours to represent different powers in the Density Spectral Array (DSA). Derived additional parameters are the Spectral Edge Frequency (SEF95), Median Frequency and the Burst Count (burst/minute) Suppression Rate (SR), Burst Suppression Ratio (BSR), Suppression Ratio (SR) or BS% Indicator (Burst suppression percentage). These can be either represented on the display by a number or visualized on the Spectrogram.

**Figure 2 jcm-10-02639-f002:**
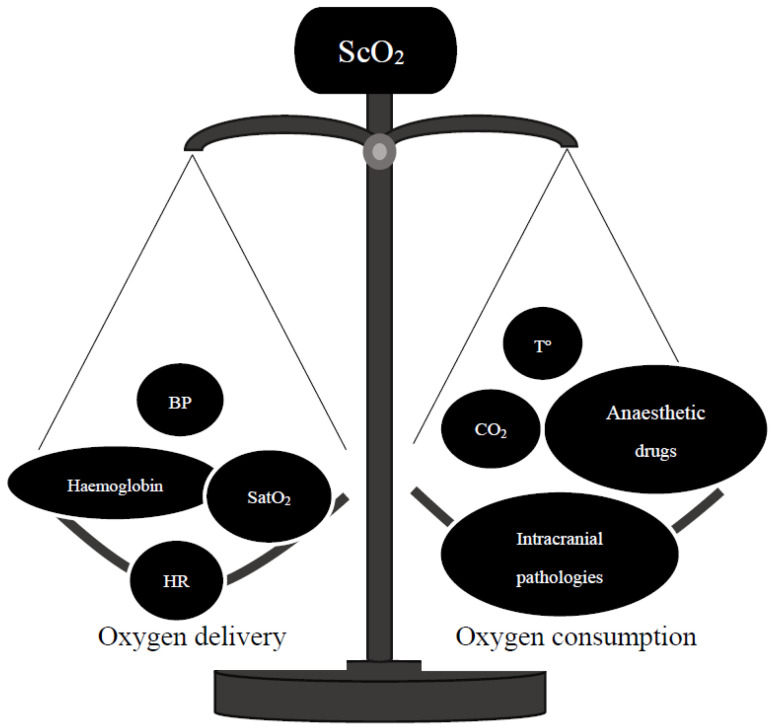
Cerebral oxygen delivery and consumption balance and contributing factors to cerebral oxygen saturation during anaesthesia. HR, heart rate; BP, blood pressure; SatO_2_, oxygen saturation; CO_2_, carbon dioxide; T°, temperature; ScO_2_ cerebral oxygen saturation.

**Table 1 jcm-10-02639-t001:** The most widely used EEG-based anaesthesia monitoring devices currently available, their key features and age limits for their safe use. EEG Electroencephalogram; NMBs Neuromuscular Blockers; EMG Electromyography; DSA Density Spectral Array; not safe = no studies demonstrate its accuracy and reliability in this age group.

Device	EEG-Derived Index (Processed EEG)	Mechanism of Action-Algorhytm Basis	Anaesthesia Range (Total Range)	Delay (s)	Potential Influencing Factors	Other EEG Monitoring Included into the Device	Additional Parameters	Index Age Limits
**BIS****Monitor** (Bispectral Index Monitor Covidien LP, Medtronic Inc.) - 2 channel - 4 channel	**BIS index**	Analysis of EEG features (degree of high frequency activation, low frequency synchronisation, nearly suppressed periods, fully suppressed periods) which correlate with hypnosis/sedations EEG in adults	40–60 (0–100) 0 = very deep hypnosis; 100 = conscious	**Index** 5–15 **DSA** Real-time	**EMG****and NMBDs** **Medical Devices** (Electocautery, pacemakers, etc.) **Certain anaesthetic agentis or adjuvants**, **Serious clinical conditions** (cerebral ischemia, hypo perfusion, cardiac arrest, hypovolemia, hypotension, hypothermia)	2 channel M: **Raw EEG** 4 channel M: **- Raw-EEG** **- DSA** (Density Spectral Array) - Asymmetry **(ASYM)** Indicator	**EMG** (bar 0–4 −> EMG power 30 to > 55) **Burst Count** (Burst/minute) **SQI** (Signal Quality Indicator) 0–100 Suppression Ratio (**SR**) number	**12 year** = safe **1-5 year** to be cautiously used (2 channel sensor age > 4 years recommended; **<1** **year** not safe **DSA** > **6 months**
**SEDLine** (Root Masimo)	**PSI** (Patient State Index)	Quantitative EEG analysis of the power within the α, β, δ and θ frequency bands; the temporal and spatial gradients occurring among these frequency bands when changing anaesthetic dept.	25–50 (0–100) 0 = very deep hypnosis; 100 = conscious	**Index** 25 **DSA** Real-time	**EMG and NMBDs** **Medical Devices** (Pacemakers, etc.) **Certain anaesthetic agentis or adjuvants**, **Serious clinical conditions** (cerebral ischemia, hypo perfusion, cardiac arrest, hypovolemia, hypotension, hypothermia)	**Raw EEG** 4 channels Power Spectrum and **DSA** **SEFL95** and SEFR95 Spectral Edge Frequency left and right (95% quantile) **ASYM**metry Graph	**EMG** Suppression Rate (**SR**) Artifact (**ARTF**) Electrical **Impedance**	<**1** **year** not safe **DSA** > **6 months**
**NARCOTREND****-Compact M** (Medival)	**Narcotrend Monitor**	Analysis of multivariate EEG-derived parameters to carry out an automatic classification of the EEG on a scale ranging from stage A (conscious) to stage F (very deep hypnosis).	D (general anaesthesia)-E (general anaesthesia with deep hypnosis); [A (conscious) − F (very deep hypnosis)].	Real-time	**EMG and NMBDs** **Medical Devices** (Electocautery, pacemakers, etc.) **Serious clinical conditions** (cerebral ischemia, hypo perfusion, cardiac arrest, hypovolemia, hypotension, hypothermia)	**Raw-EEG** (1 or 2 channels) Cerebrogram Relative Band Activities/Power Power Spectrum and **DSA** Quantiles **SEF50** (median (50% quantile) and **SEF95** spectral edge frequency (95% quantile)	**EMG** Burst Suppression Ratio (**BSR**) **STI** (sharp transient intensity) **Impedance**	**<1 year** not safe **<60 days**: only EEG classifications for stages with implied or clear suppression lines (E2 to F1). If there is an EEG without suppression lines, the output is “Undiff. EEG” **60 day–1 year**: the full A–F scale is displayed. As long as no fully classifiable EEG is detected −> the output “Undiff. EEG” is displayed. **DSA** > **6 months**
	**Narcotrend Index**		20–64 (0–100) 0 = very deep hypnosis; 100 = conscious	**Index** 28 **DSA** Real-time				
**E-****ENTROPY****Module** (GE Healthcare, Inc.)	**SE** (State of Entropy) **RE** (Response Entropy)	Analysis of the irregularity, complexity, or unpredictability characteristics of the EEG and the frontal electromyograph (FEMG) signals	40–60 (0–91)	15–60	**Medical Devices** (Electocautery, pacemakers, etc.) **Neurological Disorders**, traumas, epileptic seizures and psychoactive **medication**	–	**EMG** (Index component) Burst Suppression Ratio (**BSR**)	**<****2 year** not safe
**CS****M** Cerebral State Monitor (Danmeter)	**CSI** (Cerebral State Index)	Quantitative EEG analysis in specific frequency bands (α and β), the relationship between these quantities (β-α) and the amount of instantaneous burst suppression (BS) in each thirty-second period of the EEG.	40–60 (0–100) 0 = very deep hypnosis; 100 = conscious	50	**Serious clinical conditions** (severe neurological disorders) and psychoactive **medication** **Medical Devices** (Pacemakers)	**Raw EEG**	**EMG**% and EMG bar **BS%** Indicator (Burst suppression percentage **SQI** (Signal Quality Indicator) 0–100 Sensor **Impedance**	**<****2 year** not safe

## Data Availability

Not applicable.

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
