# Peer review of "Applications and Limitations of Neuro-Monitoring in Paediatric Anaesthesia and Intravenous Anaesthesia: A Narrative Review"

_jcm, 2021, doi:10.3390/jcm10122639_

Round 1

Reviewer 1 Report

Dear Authors

The article presented seems concise and helpful in reviewing the concepts, applications and limitations of neuromonitoring in pediatric age. However, in relation to the monitoring of cerebral autoregulation with the NIRS, I think that the paper is not adequately updated, as there is published work not mentioned, showing its added value. This particular item should be revised and possible suggestions are:

  • Measuring Near-Infrared Spectroscopy Derived Cerebral Autoregulation in Neonates: From Research Tool Toward Bedside Multimodal Monitoring
  • Cerebral Oxygenation and Autoregulation in Preterm Infants (Early NIRS Study)

  • Neonatal NIRS monitoring: recommendations for data capture and review of analytics

Author Response

Dear Reviewer

Thank you for your comments.

We are aware of the numerous studies that are recently published, but the majority of them (included the ones you mentioned above) are not related with anaesthesia. We agree that there is a lot to say about NIRS in neonates and in intensive care as well as different techniques of analysis and data collection / interpretation. However, we the updated version includes the most relevant studies of the use of NIRS during anaesthesia in children. 

Reviewer 2 Report

Thank you for permitting me to review this narrative paper 

Anstract 

There should a hint toward NIRS  which is at least part of this review 

page 1 

Please provide reference  (PPR) line 35,38

please rephrase : line 40-43

Table 1 is difficult to read please improve , howvever the contents is interesting 

when the authors say  unsafety in young children , please explain which complications are expected 

page 2  PPR  line  44-to 53  and  line 60

page 4 PPR line 1-4 and 17

page 6  PPR line 140-142

page 8  PPR line 234

The conclusion need to be shortened to a concise clinical message and perhaps a futur trend in research  

Author Response

Dear Reviewer, 

below point-by-point answer. 

page 1 Please provide reference  (PPR) line 35,38 - A: Provided

please rephrase : line 40-43 - A: Done

page 2  PPR  line  44-to 53  and  line 60 - A:Provided

page 4 PPR line 1-4 and 17 - A: Provided

page 6  PPR line 140-142 Included under reference 57-59 - A: done

page 8  PPR line 234 Included under reference 69 - A: done

The conclusion need to be shortened to a concise clinical message and perhaps a future trend in research  - A: done

Reviewer 3 Report

Dear Authors

Thank you for submitting the review article on neuromonitoring in pediatric anesthesia in the "Journal of clinical medicine".

This is a well written review article which focused on monitoring the depth of anesthesia and the other parameters to detect cerebral oxygen saturation during anesthesia for neuroprotection.

Even though there are several published articles regarding the depth of anesthesia and awareness in adult population, the literature is lagging in pediatric population.

Upon careful review of the manuscript, these are my comments

  1. The authors have described various types of EEGS for monitoring the depth of anesthesia and had shown the differences in a tabular form . Can the authors please  describe briefly about various types of EEGS with a separate side heading and if possible in a pictorial representation, so that it can be beneficial for the beginners and trainees.

2. Neurological disease: It is mentioned that caution should be employed in neurological diseases while interpreting the values, can the authors please give some examples of those diseases in pediatric population.

3. page 6: NIRS for premature babies. The authors recommend NIRS for premature babies as they have immature nervous system and auto dysregulation. Do they suggest any weight limitations ? The NIRS patches cannot be accommodated on the small forehead of a very small premature baby and lack of good contact to the skin can lead to misinterpretation or the value cannot be recorded. Does the authors have any personal experience with using NIRS in premature babies for cerebral saturation? Please clarify?

Thank you. 

Author Response

Dear Reviewer, 

below a point-by-point answer. 

1. The authors have described various types of EEGS for monitoring the depth of anesthesia and had shown the differences in a tabular form . Can the authors please  describe briefly about various types of EEGS with a separate side heading and if possible in a pictorial representation, so that it can be beneficial for the beginners and trainees.

A: We have changed the header of the chapter where we briefly describe the currently available EEG derived monitoring systems. We also added an image as suggested. 

  1. Neurological disease: It is mentioned that caution should be employed in neurological diseases while interpreting the values, can the authors please give some examples of those diseases in pediatric population.

A: In the literature there are very few studies on this topic, nonetheless EEG monitoring peculiarities have been described for both neurological diseases due to congenital metabolic and genetic disorders, and for acquired diseases such as post-hypoxic encephalopathy or neurodegenerative disorders of unknown origin. These are probably related to epileptic and non-epileptic EEG anomalies caused by the underlying neurological disorder, however the same effect may also be due to the use of drugs that act on the nervous system such as antiepileptics or neuroleptics. Nonetheless, not only the EEG monitors can be used but also they are  extremely useful in those patients whose communication problems and severe underlying neurological impairment can make it difficult to assess an adequate level of anaesthesia. We implemented the related chapter. 

  1. page 6: NIRS for premature babies. The authors recommend NIRS for premature babies as they have immature nervous system and auto dysregulation. Do they suggest any weight limitations ? The NIRS patches cannot be accommodated on the small forehead of a very small premature baby and lack of good contact to the skin can lead to misinterpretation or the value cannot be recorded. Does the authors have any personal experience with using NIRS in premature babies for cerebral saturation? Please clarify?

A: In the literature, there are no limits in terms of NIRS use and weight of preterm babies. Clinical decision in these circumstances is based on common sense. Often only one patch is applied and this is enough to generate an accurate ScO2 reading. Despite the limitations related to the technical issues, cerebral NIRS is routinely and successfully used in many neonatal intensive care units. We agree that these limitations need to be known to avoid misinterpretation of NIRS values but the importance of this monitoring seems to overcome the technical difficulties.